# Tartan: Accelerating Fully-Connected and Convolutional Layers in Deep Learning Networks by Exploiting Numerical Precision Variability

**Alberto Delmás Lascorz, Sayeh Sharify, Patrick Judd & Andreas Moshovos**
Electrical and Computer Engineering
University of Toronto
Toronto, ON, M5S 3G4, Canada
{delmasl1,sayeh,judd,moshovos}@ece.utoronto.ca

## Abstract

*Tartan TRT* a hardware accelerator for inference with Deep Neural Networks (DNNs) is presented and evaluated on Convolutional Neural Networks. *TRT* exploits the variable per layer precision requirements of DNNs to deliver execution time that is proportional to the precision $p$ in bits used per layer for convolutional and fully-connected layers. Prior art has demonstrated an accelerator with the same execution performance only for convolutional layersJudd et al. (2016a;c). Experiments on image classification CNNs show that on average across all networks studied, *TRT* outperforms a state-of-the-art bit-parallel accelerator Chen et al. (2014b) by $1.90\times$ without any loss in accuracy while it is $1.17\times$ more energy efficient. *TRT* requires no network retraining while it enables trading off accuracy for additional improvements in execution performance and energy efficiency. For example, if a 1% relative loss in accuracy is acceptable, *TRT* is on average $2.04\times$ faster and $1.25\times$ more energy efficient than a conventional bit-parallel accelerator. A Tartan configuration that processes 2-bits at time, requires less area than the 1-bit configuration, improves efficiency to $1.24\times$ over the bit-parallel baseline while being 73% faster for convolutional layers and 60% faster for fully-connected layers is also presented.

## 1 Introduction

It is only recently that commodity computing hardware in the form of graphics processors delivered the performance necessary for practical, large scale Deep Neural Network applications Krizhevsky et al. (2012). At the same time, the end of Dennard Scaling in semiconductor technology Esmaeilzadeh et al. (2011) makes it difficult to deliver further advances in hardware performance using existing general purpose designs. It seems that further advances in DNN sophistication would have to rely mostly on algorithmic and in general innovations at the software level which can be helped by innovations in hardware design. Accordingly, hardware DNN accelerators have emerged. The DianNao accelerator family was the first to use a wide single-instruction single-data (SISD) architecture to process up to 4K operations in parallel on a single chip Chen et al. (2014a;b) outperforming graphics processors by two orders of magnitude. Development in hardware accelerators has since proceeded in two directions: either toward more general purpose accelerators that can support more machine learning algorithms while keeping performance mostly on par with DaDianNao (*DaDN*) Chen et al. (2014b), or toward further specialization of specific layers or classes of DNNs with the goal of outperforming *DaDN* in execution time and/or energy efficiency, e.g., Han et al. (2016); Albericio et al. (2016a); Judd et al. (2016a); Chen, Yu-Hsin and Krishna, Tushar and Emer, Joel and Sze, Vivienne (2016); Reagen et al. (2016). This work is along the second direction. Section 5 reviews several other accelerator designs.

While *DaDN*'s functional units process 16-bit fixed-point values, DNNs exhibit varying precision requirements across and within layers, e.g., Judd et al. (2015). Accordingly, it is possible to use

shorter, per layer representations for activations and/or weights. However, with existing bit-parallel functional units doing so does not translate into a performance nor an energy advantage as the values are expanded into the native hardware precision inside the unit.

This work presents *Tartan* (*TRT*), a massively parallel hardware accelerator whose execution time for fully-connected and convolutional layers scales with the precision $p$ used to represent the input values. *TRT* uses hybrid bit-serial/bit-parallel functional units and exploits the abundant parallelism of typical DNN layers with the goal of exceeding *DaDN*'s execution time performance and energy efficiency. Ideally *Tartan* can improve execution time by $\frac{16}{p}$ where $p$ is the precision used for the activations in convolutional layers, and for the activations and weights in fully-connected layers. Every bit of precision that can be eliminated ideally reduces execution time and increases energy efficiency. *TRT* builds upon the *Stripes* (*STR*) accelerator Judd et al. (2016c;a) which improves execution time and energy efficiency only for convolutional layers.

This work evaluates *TRT* on a set of convolutional neural networks (CNNs) for image classification. On average *TRT* reduces inference time by $1.61\times$, $1.91\times$ and $1.90\times$ over *DaDN* for the fully-connected, the convolutional, and all layers respectively. Energy efficiency compared to *DaDN* with *TRT* is $0.92\times$, $1.18\times$ and $1.17\times$ respectively. *TRT* enables trading off accuracy for improving execution time and energy efficiency. For example, on average for the fully-connected layers, accepting a 1% loss in accuracy improves performance to $1.73\times$ and energy efficiency to $1.00\times$ compared to *DaDN*.

The rest of this document is organized as follows: Section 2 illustrates the key concepts behind *TRT* via an example. Section 3 reviews the *DaDN* architecture and presents an equivalent *Tartan* configuration. Section 4 presents the experimental results. Section 5 reviews related work and discusses the limitations of this study and the potential challenges with *TRT*. Section 6 concludes.

## 2  *Tartan*: A Simplified Example

This section illustrates at a high-level the *TRT* design by showing how it would process two purposely trivial cases: 1) a fully-connected layer (FCL) with a single input activation producing two output activations, and 2) a convolutional layer (CVL) with two input activations and one single-weight filter producing two output activations. The per layer calculations are:

$$Fully - Connected: \qquad\qquad Convolutional:$$
$$f_1 = w_1 \times a \qquad\qquad c_1 = w \times a_1$$
$$f_2 = w_2 \times a \qquad\qquad c_2 = w \times a_2$$

Where $f_1$, $f_2$, $c_1$ and $c_2$ are output activations, $w_1$, $w_2$, and $w$ are weights, and $a_1$, $a_2$ and $a$ are input activations. For clarity all values are assumed to be represented in 2 bits of precision.

### 2.1  Conventional Bit-Parallel Processing

Figure 2.1a shows a bit-parallel processing engine representative of *DaDN*. Every cycle, the engine can calculate the product of two 2-bit inputs, $i$ (weight) and $v$ (activation) and accumulate or store it into the output register $OR$. Parts (b) and (c) of the figure show how this unit can calculate the example CVL over two cycles. In part (b) and during cycle 0, the unit accepts along the $v$ input bits 0 and 1 of $a_1$ (noted as $a1/0$ and $a1/1$ respectively on the figure), and along $i$ bits 0 and 1 of $w$ and produces both bits of output $c_1$. Similarly, during cycle 1 (part (c)), the unit processes $a_2$ and $w$ to produce $c_2$. In total, over two cycles, the engine produced two $2b \times 2b$ products. Processing the example FCL also takes two cycles: In the first cycle $w_1$ and $a$ produce $f_1$, and in the second cycle $w_2$ and $a$ produce $f_2$. This process is not shown in the interest of space.

### 2.2  *Tartan*'s Approach

Figure 2 shows how a *TRT*-like engine would process the example CVL. Figure 2a shows the engine's structure which comprises two subunits. The two subunits accept each one bit of an activation per cycle through inputs $v0$ and $v1$ respectively and as before, there is a common 2-bit weight input $(i1, i0)$. In total, the number of input bits is 4, identical to the bit-parallel engine.

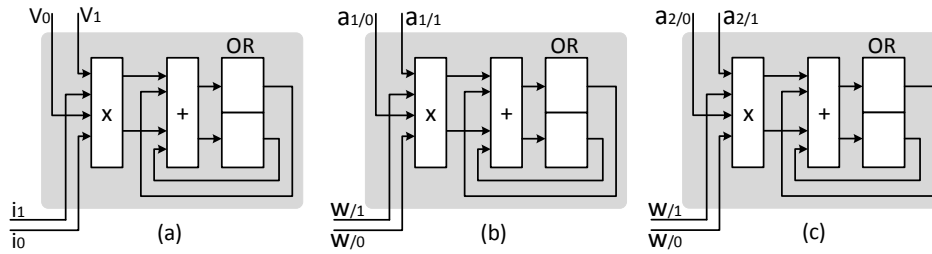

Figure 1: Bit-Parallel Engine processing the convolutional layer over two cycles: a) Structure, b) Cycle 0, and c) Cycle 1.

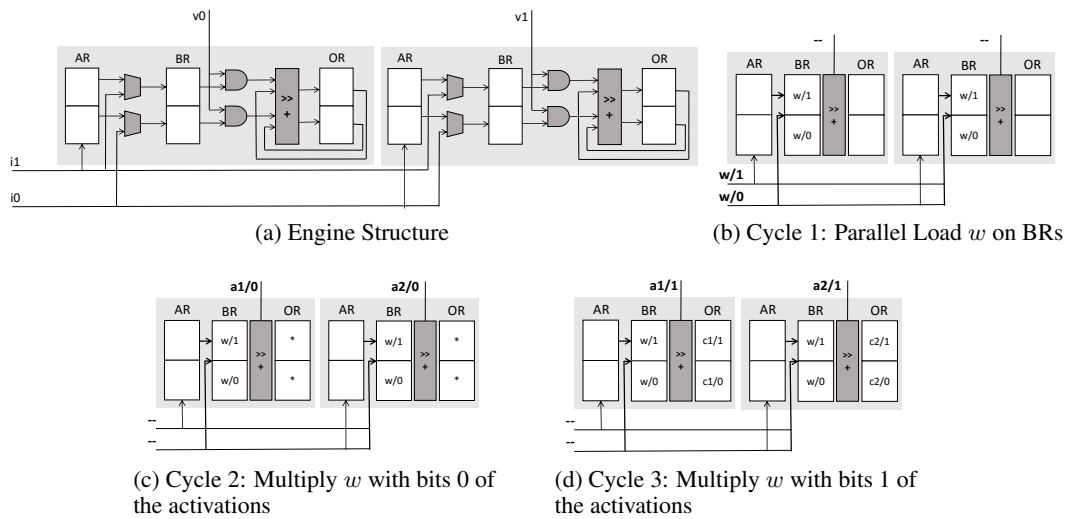

(a) Engine Structure

(b) Cycle 1: Parallel Load $w$ on BRs

(c) Cycle 2: Multiply $w$ with bits 0 of the activations

(d) Cycle 3: Multiply $w$ with bits 1 of the activations

Figure 2: Processing the example Convolutional Layer Using *TRT*'s Approach.

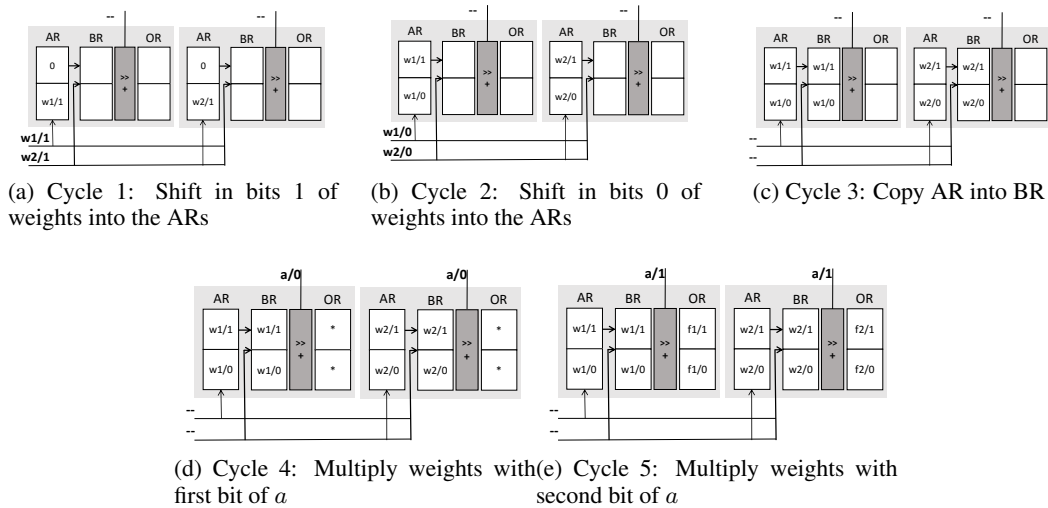

(a) Cycle 1: Shift in bits 1 of weights into the ARs

(b) Cycle 2: Shift in bits 0 of weights into the ARs

(c) Cycle 3: Copy AR into BR

(d) Cycle 4: Multiply weights with first bit of $a$

(e) Cycle 5: Multiply weights with second bit of $a$

Figure 3: Processing the example Fully-Connected Layer using *TRT*'s Approach.

Each subunit contains three 2-bit registers: a shift-register AR, a parallel load register BR, and an parallel load output register OR. Each cycle each subunit can calculate the product of its single bit $v_i$ input with BR which it can write or accumulate into its OR. There is no bit-parallel multiplier since the subunits process a single activation bit per cycle. Instead, two AND gates, a shift-and-add functional unit, and OR form a shift-and-add multiplier/accumulator. Each AR can load a single bit per cycle from one of the $i$ wires, and BR can be parallel loaded from AR or from the $i$ wires.

**Convolutional Layer:** Figure 2b through Figure 2d show how the CVL is processed. The figures abstract away the unit details showing only the register contents. As Figure 2b shows, during cycle 1, the $w$ synapse is loaded in parallel to the BRs of both subunits via the $i1$ and $i0$ inputs. During cycle 2, bits 0 of $a_1$ and of $a_2$ are sent via the $v0$ and $v1$ inputs respectively to the first and second subunit. The subunits calculate concurrently $a_1/0 \times w$ and $a_2/0 \times w$ and accumulate these results into their ORs. Finally, in cycle 3, bit 1 of $a_1$ and $a_2$ appear respectively on $v0$ and $v1$. The subunits calculate respectively $a_1/1 \times w$ and $a_2/1 \times w$ accumulating the final output activations $c_1$ and $c_2$ into their ORs.

In total it took 3 cycles to process the layer. However, at the end of the third cycle, another $w$ could have been loaded into the BRs (the $i$ are idle) allowing a new set of outputs to commence computation during cycle 4. That is loading a new weight can be hidden during the processing of the current output activation for all but the first time. In the steady state, when the input activations are represented in two bits, this engine will be producing two $2b \times 2b$ terms every two cycles thus matching the bandwidth of the bit-parallel engine.

If the activations $a_1$ and $a_2$ could be represented in just one bit, then this engine would be producing two output activations per cycle, twice the bandwidth of the bit-parallel engine. The latter is incapable of exploiting the reduced precision. In general, if the bit-parallel hardware was using $P_{base}$ bits to represent the activations while only $P_a$ bits were enough, *TRT* would outperform the bit-parallel engine by $\frac{P_{base}}{P_{TRT}}$.

**Fully-Connected Layer:** Figure 3 shows how a *TRT*-like unit would process the example FCL. As Figure 3a shows, in **cycle 1**, bit 1 of $w_1$ and of $w_2$ appear respectively on lines $i1$ and $i0$. The left subunit's AR is connected to $i1$ while the right subunit's AR is connected to $i0$. The ARs shift in the corresponding bits into their least significant bit sign-extending to the vacant position (shown as a 0 bit on the example). During **cycle 2**, as Figure 3b shows, bits 0 of $w1$ and of $w2$ appear on the respective $i$ lines and the respective ARs shift them in. At the end of the cycle, the left subunit's AR contains the full 2-bit $w_1$ and the right subunit's AR the full 2-bit $w_2$. In **cycle 3**, Figure 3c shows that the contents of AR are copied to BR in each subunit. From the next cycle, calculating the products can now proceed similarly to what was done for the CVL. In this case, however, each BR contains a different weight whereas in the CVL all BRs held the same $w$ value. The shift capability of the ARs coupled with the different $i$ wire per subunit connection allowed us to load a different weight bit-serially over two cycles. Figure 3d and Figure 3e show cycles 4 and 5 respectively. During **cycle 4**, bit 0 of $a1$ appears on both $v$ inputs and is multiplied with the BR in each subunit. In **cycle 5**, bit 1 of $a1$ appears on both $v$ inputs and the subunits complete the calculation of $f_1$ and $f_2$. It takes two cycles to produce the two $2b \times 2b$ products once the correct inputs appear into the BRs.

While in our example no additional inputs nor outputs are shown, it would have been possible to overlap the loading of a new set of $w$ inputs into the ARs while processing the current weights stored into the BRs. That is the loading into ARs, copying into BRs, and the bit-serial multiplication of the BRs with the activations is a 3-stage pipeline where each stage can take multiple cycles. In general, assuming that both activations and weights are represented using 2 bits, this engine would match the performance of the bit-parallel engine in the steady state. When both set of inputs $i$ and $v$ can be represented with fewer bits, 1 in this case, the engine would produce two terms per cycle, twice the bandwidth of the bit-parallel engine of the previous section.

**Summary:** In general, if $P_{base}$ the precision of the bit-parallel engine, and $P_a^L$ and $P_w^L$ the precisions that can be used respectively for activations and weights for layer $L$, a *TRT* engine can ideally outperform an equivalent bit parallel engine by $\frac{P_{base}}{P_a^L}$ for CVLs, and by $\frac{P_{base}}{max(P_a^L, P_w^L)}$ for FCLs. This example used the simplest *TRT* engine configuration. Since typical layers exhibit massive parallelism, *TRT* can be configured with many more subunits while exploiting weight reuse for CVLs and activation reuse for FCLs. The next section describes the baseline state-of-the-art DNNs accelerator and presents an equivalent *TRT* configuration.

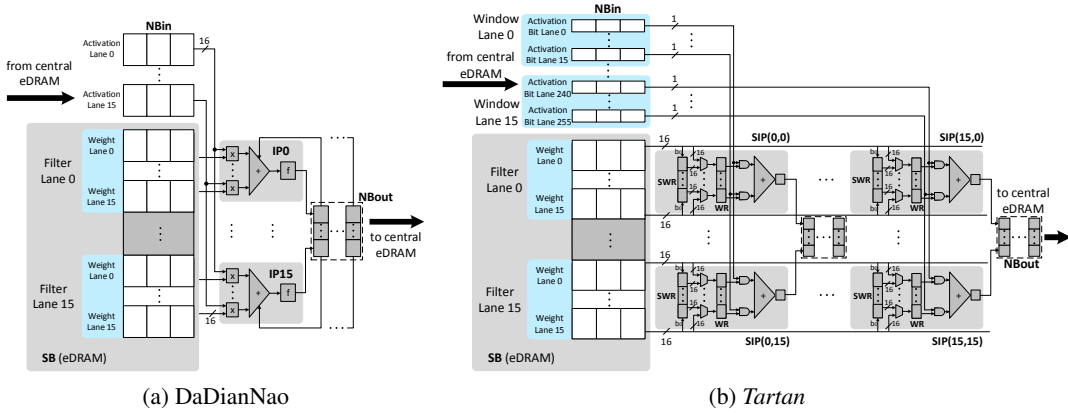

Figure 4: Processing Titles.

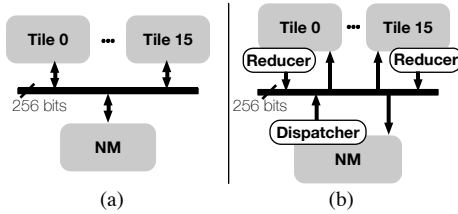

Figure 5: Overview of the system components and their communication. a) *DaDN*. b) *Tartan*.

# 3 *Tartan* ARCHITECTURE

This work presents *TRT* as a modification of the state-of-the-art *DaDianNao* accelerator. Accordingly, Section 3.1 reviews *DaDN*'s design and how it can process FCLs and CVLs. For clarity, in what follows the term *brick* refers to a set of 16 elements of a 3D activation or weight array[1] input which are contiguous along the $i$ dimension, e.g., $a(x, y, i)...a(x, y, i + 15)$. Bricks will be denoted by their origin element with a $B$ subscript, e.g., $a_B(x, y, i)$. The size of a brick is a design parameter.

## 3.1 BASELINE SYSTEM: DADIANNAO

*TRT* is demonstrated as a modification of the *DaDianNao* accelerator (*DaDN*) proposed by Chen et al. (2014b). Figure 4a shows a *DaDN* tile which processes 16 filters concurrently calculating 16 activation and weight products per filter for a total of 256 products per cycle. Each cycle the tile accepts 16 weights per filter for total of 256 synapses and 16 input activations. The tile multiplies each weight with only one activation whereas each activation is multiplied with 16 weights, one per filter. The tile reduces the 16 products into a single partial output activation per filter, for a total of 16 partial output activations for the tile. Each *DaDN* chip comprises 16 such tiles, each processing a different set of 16 filters per cycle. Accordingly, each cycle, the whole chip processes 16 activations and $256 \times 16 = 4K$ weights producing $16 \times 16 = 256$ partial output activations, 16 per tile.

Internally, each tile has: 1) a synapse buffer (SB) that provides 256 weights per cycle one per weight lane, 2) an input neuron buffer (NBin) which provides 16 activations per cycle through 16 neuron lanes, and 3) a neuron output buffer (NBout) which accepts 16 partial output activations per cycle. In the tile's datapath each activation lane is paired with 16 weight lanes one from each filter. Each synapse and neuron lane pair feeds a multiplier, and an adder tree per filter lane reduces the 16 per filter products into a partial sum. In all, the filter lanes produce each a partial sum per cycle, for a

---

[1]An FCL can be thought of as a CVL where the input activation array has unit x and y dimensions, and there are as many filters as output activations, and where the filter dimenions are identical to the input activation array.

total of 16 partial output activations per Once a full window is processed, the 16 resulting sums, are fed through a non-linear activation function, $f$, to produce the 16 final output activations. The multiplications and reductions needed per cycle are implemented via 256 multipliers one per weight lane and sixteen 17-input (16 products plus the partial sum from NBout) adder trees one per filter lane.

Figure 5a shows an overview of the *DaDN* chip. There are 16 processing tiles connected via an interconnect to a shared central eDRAM *Neuron Memory* (NM). *DaDN*'s main goal was minimizing off-chip bandwidth while maximizing on-chip compute utilization. To avoid fetching weights from off-chip, *DaDN* uses a 2MB eDRAM Synapse Buffer (SB) for weights per tile for a total of 32MB eDRAM. All inter-layer activation outputs except for the initial input and the final output are stored in NM which is connected via a broadcast interconnect to the 16 Input Neuron Buffers (NBin) buffers. All values are 16-bit fixed-point, hence a 256-bit wide interconnect can broadcast a full activation brick in one step. Off-chip accesses are needed only for reading: 1) the input image, 2) the weight once per layer, and 3) for writing the final output.

Processing starts by reading from external memory the first layer's filter weights, and the input image. The weights are distributed over the SBs and the input is stored into NM. Each cycle an input activation brick is broadcast to all units. Each units reads 16 weight bricks from its SB and produces a partial output activation brick which it stores in its NBout. Once computed, the output activations are stored through NBout to NM and then fed back through the NBins when processing the next layer. Loading the next set of weights from external memory can be overlapped with the processing of the current layer as necessary.

### 3.2 *Tartan*

As Section 2 explained, *TRT* processes activations bit-serially multiplying a single activation bit with a full weight per cycle. Each *DaDN* tile multiplies 16 16-bit activations with 256 weights each cycle. To match *DaDN*'s computation bandwidth, *TRT* needs to multiply 256 1-bit activations with 256 weights per cycle. Figure 4b shows the *TRT* tile. It comprises 256 Serial Inner-Product Units (SIPs) organized in a $16 \times 16$ grid. Similar to *DaDN* each SIP multiplies 16 weights with 16 activations and reduces these products into a partial output activation. Unlike *DaDN*, each SIP accepts 16 single-bit activation inputs. Each SIP has two registers, each a vector of 16 16-bit subregisters: 1) the *Serial Weight Register* (SWR), and 2) the *Weight Register* (WR). These correspond to AR and BR of the example of Section 2. NBout remains as in *DaDN*, however, it is distributed along the SIPs as shown.

**Convolutional Layers:** Processing starts by reading in parallel 256 weights from the SB as in *DaDN*, and loading the 16 per SIP row weights in parallel to all SWRs in the row. Over the next $P_a^L$ cycles, the weights are multiplied by the bits of an input activation brick per column. *TRT* exploits weight reuse across 16 windows sending a different input activation brick to each column. For example, for a CVL with a stride of 4 a *TRT* tile will processes 16 activation bricks $a_B(x,y,i)$, $a_B(x+4,y,i)$ through $a_B(x+63,y,i)$ in parallel a bit per cycle. Assuming that the tile processes filters $f_i$ though $f_{i+15}$, after $P_a^L$ cycles it would produce the following *partial* output activations: $o_B(x/4,y/4,f_i)$, through $o_B(x/4+15,y/4,f_i)$, that is 16 contiguous on the $x$ dimension output activation bricks. Whereas *DaDN* would process 16 activations bricks over 16 cycles, *TRT* processes them concurrently but bit-serially over $P_a^L$ cycles. If $P_a^L$ is less than 16, *TRT* will outperform *DaDN* by $16/P_a^L$, and when $P_a^L$ is 16, *TRT* will match *DaDN*'s performance.

**Fully-Connected Layers:** Processing starts by loading bit-serially and in parallel over $P_w^L$ cycles, 4K weights into the SWRs. Each SWR per row gets a different set of 16 weights as each subregister is connected to one out of the 256 wires of the SB output bus for the SIP row. Once the weights have been loaded, the SWRs are copied to the SWs and multiplication with the input activations can then proceed bit-serially over $P_a^L$ cycles. Assuming that there are enough output activations so that a different output activation can be assigned to each SIP, the same input activation brick can be broadcast to all SIP columns. For example, for an FCL a *TRT* tile will process one activation brick $a_B(i)$ bit-serially to produce 16 output activation bricks $o_B(i)$ through $o_B(i \times 16)$ one per SIP column. Loading the next set of weights can be done in parallel with processing the current set, thus execution time is constrained by $P_{max}^L = max(P_a^L, P_w^L)$. Thus, a *TRT* tile produces 256 partial

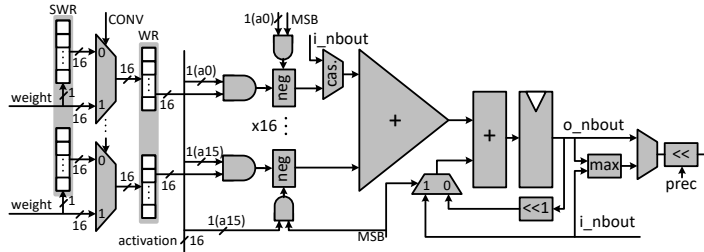

Figure 6: *TRT*'s SIP.

output activations every $P_{max}^L$ cycles, a speedup of $16/P_{max}$ over *DaDN* since a *DaDN* tile always needs 16 cycles to do the same.

For *TRT* to be fully utilized an FCL must have at least 4K output activations. Some of the networks studied have a layer with as little as 2K output activations. To avoid underutilization, the SIPs along each row are cascaded into a daisy-chain, where the output of one can feed into an input of the next via a multiplexer. This way, the computation of an output activation can be sliced over the SIPs along the same row. In this case, each SIP processes only a portion of the input activations resulting into several partial output activations along the SIPs on the same row. Over the next $np$ cycles, where $np$ the number of slices used, the $np$ partial outputs can be reduced into the final output activation. The user can chose any number of slices up to 16, so that *TRT* can be fully utilized even with fully-connected layers of just 256 outputs. For example, in NeuralTalk Karpathy & Li (2014) the smallest layers can have 600 outputs or fewer.

**Other Layers:** *TRT* like *DaDN* can process the additional layers needed by the studied networks. For this purpose the tile includes additional hardware support for max pooling similar to *DaDN*. An activation function unit is present at the output of NBout in order to apply nonlinear activations before the output neurons are written back to NM.

### 3.3 SIP AND OTHER COMPONENTS

**SIP: Bit-Serial Inner-Product Units:** Figure 6 shows *TRT*'s Bit-Serial Inner-Product Unit (SIP). Each SIP multiplies 16 activations by 16 weights to produce an output activation. Each SIP has two registers, a Serial Weight Register (SWR) and a Weight Registers (WR), each containing 16 16-bit subregisters. Each SWR subregister is a shift register with a single bit connection to one of the weight bus wires that is used to read weights bit-serially for FCLs. Each WR subregister can be parallel loaded from either the weight bus or the corresponding SWR subregister, to process CVLs or FCLs respectively. Each SIP includes 256 2-input AND gates that multiply the weights in the WR with the incoming activation bits, and a $16 \times 16b$ adder tree that sums the partial products. A final adder plus a shifter accumulate the adder tree results into an output register. In each SIP, a multiplexer at the first input of the adder tree implements the cascade mode supporting slicing the output activation computation along the SIPs of a single row. To support signed 2's complement neurons, the SIP can subtract the weight corresponding to the most significant bit (MSB) from the partial sum when the MSB is 1. This is done with negation blocks for each weight before the adder tree. Each SIP also includes a comparator (max) to support max pooling layers.

**Dispatcher and Reducers:** Figure 5b shows an overview of the full *TRT* system. As in *DaDN* there is a central NM and 16 tiles. A *Dispatcher* unit is tasked with reading input activations from NM always performing eDRAM-friendly wide accesses. It transposes each activation and communicates each a bit a time over the global interconnect. For CVLs the dispatcher has to maintain a pool of multiple activation bricks, each from different window, which may require fetching multiple rows from NM. However, since a new set of windows is only needed every $P_a^L$ cycles, the dispatcher can keep up for the layers studied. For FCLs one activation brick is sufficient. A *Reducer* per title is tasked with collecting the output activations and writing them to NM. Since output activations take multiple cycles to produce, there is sufficient bandwidth to sustain all 16 tiles.

### 3.4 PROCESSING SEVERAL BITS AT ONCE

In order to improve *TRT*'s area and power efficiency, the number of bits processed at once can be parameterized. In this case, the weights are multiplied with several activation bits at once, and the multiplication results are partially shifted before they are inserted into their corresponding adder tree.

In order to load the weights on time, the SWR subregister has to be modified so it can load several bits in parallel, and shift that number of positions every cycle. The negation block (for 2's complement support) will operate only over the most significant product result.

The chief advantage of such a design is that less SIPs are needed in order to achieve the same throughput – for example, processing 2 bits at once allows reducing the number of columns from 16 to 8. Although the total number of bus wires is similar, the distance they have to cover is significantly reduced. Likewise, the total number of adders required stays similar, but they are clustered closer together.

A drawback of this design is the limitation to precisions that are exact multiples of the number of bits processed at once.

## 4 EVALUATION

This section evaluates *TRT*'s performance, energy and area and explores the trade-off between accuracy and performance comparing to *DaDN*.

### 4.1 METHODOLOGY

**Numerical Representation Requirements Analysis:** The per layer precision profiles are found via the methodology of Judd et al. Judd et al. (2015). Caffe Jia et al. (2014) was used to measure how reducing the precision of each FCL affects the network's overall *top-1* prediction accuracy over 5000 images. The network definitions and pre-trained synaptic weights are taken from the Caffe Model Zoo Jia (2015). Since *TRT*'s performance for FCLs is bound by the maximum of the weight and activation precisions, our exploration was limited to the cases where both are the same. The search procedure is a gradient descent where a given layer's precision is iteratively decremented one bit at a time, until the network's accuracy drops. For weights, the fixed point numbers are set to represent values between -1 and 1. For activations, the number of fractional bits is fixed to a previously-determined value known not to hurt accuracy, as per Judd et al. (2015). While both activations and weights use the same number of bits, their precisions and ranges differ.

**Performance, Area and Energy:** *DaDN*, *STR* and *TRT* were modeled using the same methodology for consistency. A custom cycle-accurate simulator models execution time. Computation was scheduled as described by Judd et al. (2016a) to maximize energy efficiency for *DaDN*. The logic components of the both systems were synthesized with the Synopsys Design Compiler Synopsys for a TSMC 65nm library to report power and area. The circuit is clocked at 980 MHz. The NBin and NBout SRAM buffers were modelled using CACTI Muralimanohar & Balasubramonian. The eDRAM area and energy were modelled with *Destiny* Poremba et al. (2015).

### 4.2 RESULTS

**Fully-Connected Layer Precisions:** Table 1 reports the per layer precisions for the CVLs and FCLs of the networks studied along with the speedup over *DaDN* that would be ideally possible. The discussion in this section focuses solely on FCLs. The precisions that can be used vary from 8 up to 10 bits vs. the 16 bits *DaDN* uses. The ideal speedup ranges from 63% to 66% with no accuracy loss. Additional exploration of the precision space may yield even shorter precisions without sacrificing accuracy. Modest additional improvements are possible with a loss of 1% in accuracy.

**Execution Time:** Table 2 reports *TRT*'s performance and energy efficiency relative to *DaDN* for the precision profiles in Table 1 separately for the fully-connected layers, for the convolutional layers,

| Network | Convolutional layers | | Fully connected layers | |
|---|---|---|---|---|
| | **Per Layer Activation Precision in Bits** | **Ideal Speedup** | **Per Layer Activation and Weight Precision in Bits** | **Ideal Speedup** |
| **100% Accuracy** | | | | |
| AlexNet | 9-8-5-5-7 | 2.38 | 10-9-9 | 1.66 |
| VGG_S | 7-8-9-7-9 | 2.04 | 10-9-9 | 1.64 |
| VGG_M | 7-7-7-8-7 | 2.23 | 10-8-8 | 1.64 |
| VGG_19 | 12-12-12-11-12-10-11-11-13-12-13-13-13-13-13-13 | 1.35 | 10-9-9 | 1.63 |
| **99% Accuracy** | | | | |
| AlexNet | 9-7-4-5-7 | 2.58 | 9-8-8 | 1.85 |
| VGG_S | 7-8-9-7-9 | 2.04 | 9-9-8 | 1.79 |
| VGG_M | 6-8-7-7-7 | 2.34 | 9-8-8 | 1.80 |
| VGG_19 | 9-9-9-8-12-10-10-12-13-11-12-13-13-13-13-13 | 1.57 | 10-9-8 | 1.63 |

Table 1: Per layer synapse precision profiles needed to maintain the same accuracy as in the baseline. *Ideal*: Potential speedup with *TRT* over a 16-bit bit-parallel baseline.

| Accuracy | Fully Connected Layers | | | | Convolutional Layers | | | |
|---|---|---|---|---|---|---|---|---|
| | **100%** | | **99%** | | **100%** | | **99%** | |
| | **Perf** | **Eff** | **Perf** | **Eff** | **Perf** | **Eff** | **Perf** | **Eff** |
| AlexNet | 1.61 | 0.92 | 1.80 | 1.04 | 2.32 | 1.43 | 2.52 | 1.55 |
| VGG_S | 1.61 | 0.92 | 1.76 | 1.01 | 1.97 | 1.21 | 1.97 | 1.21 |
| VGG_M | 1.61 | 0.93 | 1.77 | 1.02 | 2.18 | 1.34 | 2.29 | 1.40 |
| VGG_19 | 1.60 | 0.92 | 1.61 | 0.93 | 1.35 | 0.83 | 1.56 | 0.96 |
| **geomean** | 1.61 | 0.92 | 1.73 | 1.00 | 1.91 | 1.18 | 2.05 | 1.26 |

Table 2: Execution time and energy efficiency improvement with *TRT* compared to *DaDN*.

and the whole network. For the 100% profile, where no accuracy is lost, *TRT* yields, on average, a speedup of $1.61\times$ over *DaDN* on FCLs. With the 99% profile, it improves to $1.73\times$.

There are two main reasons the ideal speedup can't be reached in practice: dispatch overhead and underutilization. Dispatch overhead occurs on the initial $P_w^L$ cycles of execution, where the serial weight loading process prevents any useful products to be performed. In practice, this overhead is less than 2% for any given network, although it can be as high as 6% for the smallest layers. Underutilization can happen when the number of output neurons is not a power of two, or lower than 256. The last classifier layers of networks designed towards recognition of ImageNet (Russakovsky et al. (2014)) categories all provide 1000 output neurons, which leads to 2.3% of the SIPs being idle.

We have also performed an evaluation of NeuralTalk LSTM Karpathy & Li (2014) which uses long short-term memory to automatically generate image captions. Precision can be reduced down to 11 bits withouth affecting the accuracy of the predictions (measured as the BLEU score when compared to the ground truth) resulting in a ideal performance improvement of $1.45\times$ translating into a $1.38\times$ speedup with *TRT*.

**Energy Efficiency:** This section compares the *energy efficiency* or simply efficiency of *TRT* and *DaDN*. Energy Efficiency is the inverse of the relative energy consumption of the two designs. The average efficiency improvement with *TRT* across all networks and layers for the 100% profile is $1.17\times$. In the FCLs, *TRT* is not as efficient as *DaDN*, however, the energy efficiency for CVLs more than compensates when whole networks are considered except for VGG_19. Regardless, performance would not scale linearly if *DaDN* was to include more tiles in an attempt to match *TRT*'s performance: under-utilization for most layers in these networks would severely reduce any performance improvements delivered via additional tiles under *DaDN*. Overall, efficiency primarily comes from the reduction in effective computation following the use of reduced precision arithmetic for the inner product operations. Furthermore, the amount of data that has to be transmitted from the SB and the traffic between the central eDRAM and the SIPs is decreased proportionally to the chosen

| | **TRT area** $(mm^2)$ | **TRT 2-bit area** $(mm^2)$ | **DaDN area** $(mm^2)$ |
|---|---|---|---|
| **Inner-Product Units** | 57.27 (47.71%) | 37.66 (37.50%) | 17.85 (22.20%) |
| **Synapse Buffer** | 48.11 (40.08%) | 48.11 (47.90%) | 48.11 (59.83%) |
| **Input Neuron Buffer** | 3.66 (3.05%) | 3.66 (3.64%) | 3.66 (4.55%) |
| **Output Neuron Buffer** | 3.66 (3.05%) | 3.66 (3.64%) | 3.66 (4.55%) |
| **Neuron Memory** | 7.13 (5.94%) | 7.13 (7.10%) | 7.13 (8.87%) |
| **Dispatcher** | 0.21 (0.17%) | 0.21 (0.21%) | - |
| **Total** | 120.04 (100%) | 100.43 (100%) | 80.41 (100%) |
| **Normalized Total** | 1.49× | 1.25× | 1.00× |

Table 3: Area Breakdown for *TRT* and *DaDN*

| | **Fully Connected Layers** | | **Convolutional Layers** | |
|---|---|---|---|---|
| | **vs. DaDN** | **vs. 1b TRT** | **vs. DaDN** | **vs. 1b TRT** |
| AlexNet | +58% | -2.06% | +208% | -11.71% |
| VGG_S | +59% | -1.25% | +76% | -12.09% |
| VGG_M | +63% | +1.12% | +91% | -13.78% |
| VGG_19 | +59% | -0.97% | +29% | -4.11% |
| **geomean** | +60% | -0.78% | +73% | -10.36% |

Table 4: Relative performance of 2-bit *TRT* variation compared to *DaDN* and the 1-bit *TRT*

precision. When the per layer precisions are reduced adequately *TRT* becomes more efficient than *DaDN*.

**Area Overhead:** Table 3 reports the area breakdown of *TRT* and *DaDN*. Over the full chip, *TRT* needs 1.49× the area compared to *DaDN* while delivering on average a 1.90× improvement in speed. Generally, performance would scale sublinearly with area for *DaDN* due to underutilization. The 2-bit variant, which has a lower area overhead, is described in detail in the next section.

### 4.3 Two-Bit at Once Performance Evaluation

We evaluate the performance for a multi-bit design as described in section 3.4, where 2 bits are processed every cycle in as half as many total SIPs. The precisions used are the same as indicated in Table 1 for 100% accuracy, rounded up to the next multiple of two. The results are shown in Table 4. The 2-bit *TRT* always improves performance compared to *DaDN* as the "vs. *DaDN*" columns show. Compared to the 1-bit *TRT* performance is slightly lower however given that the area of the 2-bit *TRT* is much lower, this can be a good trade-off. Overall, there are two forces at work that shape performance relative to the 1-bit *TRT*. There is performance potential lost due to rounding all precisions to an even number, and there is performance benefit by requiring less parallelism. The time needed to serially load the first bundle of weights is also reduced. In VGG_19 the performance benefit due to the lower parallelism requirement outweights the performance loss due to precision rounding. In all other cases, the reverse is true.

A hardware synthesis and layout of both *DaDN* and *TRT*'s 2-bit variant using TSMC 65nm typical case libraries shows that the total area overhead can be as low as 24.9%, with an improved energy efficiency in fully connected layers of 1.24× on average.

## 5 Related Work and Limitations of this Work

The recent success of Deep Learning has led to several proposals for hardware acceleration of DNNs. This section reviews some of these recent efforts. However, specialized hardware designs for neural networks is a field with a relatively long history. Relevant to *TRT*, bit-serial processing hardware for neural networks has been proposed several decades ago, e.g., Svensson & Nordstrom (1990); Murray et al. (1988). While the performance of these designs scales with precision it would be lower than that of an equivalently configured bit-parallel engine. For example, Svensson & Nordstrom (1990) uses an interesting bit-serial multiplier which requires $O(4 \times p)$ cycles, where $p$ the precision in bits. Furthermore, as semiconductor technology has progressed the number of resources that can be

put on chip and the trade offs (e.g., relative speed of memory vs. transistors vs. wires) are today vastly different facilitating different designs. However, truly bit-serial processing such as that used in the aforementioned proposals needs to be revisited with today's technology constraints due to its potentially high compute density (compute bandwidth delivered per area).

In general, hardware acceleration for DNNs has recently progressed in two directions: 1) considering more general purpose accelerators that can support additional machine learing algorithms, and 2) considering further improvements primarily for convolutional neural networks and the two most dominant in terms of execution time layer types: convolutional and fully-connected. In the first category there are accelerators such as Cambricon Liu et al. (2016) and Cambricon-X Zhang et al. (2016). While targeting support for more machine learning algorithms is desirable, work on further optimizing performance for specific algorithms such as *TRT* is valuable and needs to be pursued as it will affect such general purpose accelerators.

*TRT* is closely related to *Stripes* Judd et al. (2016c;a) whose execution time scales with precision but only for CVLs. *STR* does not improve performance for FCLs. *TRT* improves upon *STR* by enabling: 1) performance improvements for FCLs, and 2) slicing the activation computation across multiple SIPs thus preventing underutilization for layers with fewer than 4K outputs. *Pragmatic* uses a similar in spirit organization to *STR* but its performance on CVLs depends only on the number of activation bits that are 1 Albericio et al. (2016b). It should be possible to apply the *TRT* extensions to Pragmatic, however, performance in FCLs will still be dictated by weight precision. The area and energy overheads would need to be amortized by a commensurate performance improvement.

The *Efficient Inference Engine* (EIE) uses synapse pruning, weight compression, zero activation elimination, and network retraining to drastically reduce the amount of computation and data communication when processing fully-connected layers Han et al. (2016). An appropriately configured EIE will outperform *TRT* for FCLs, provided that the network is pruned and retrained. However, the two approaches attack a different component of FCL processing and there should be synergy between them. Specifically, EIE currently does not exploit the per layer precision variability of DNNs and relies on retraining the network. It would be interesting to study how EIE would benefit from a *TRT*-like compute engine where EIE's data compression and pruning is used to create vectors of weights and activations to be processed in parallel. EIE uses single-lane units whereas *TRT* uses a coarser-grain lane arrangement and thus would be prone to more imbalance. A middle ground may be able to offer some performance improvement while compensating for cross-lane imbalance.

Eyeriss uses a systolic array like organization and gates off computations for zero activations Chen, Yu-Hsin and Krishna, Tushar and Emer, Joel and Sze, Vivienne (2016) and targets primarily high-energy efficiency. An actual prototype has been built and is in full operation. Cnvlutin is a SIMD accelerator that skips on-the-fly ineffectual activations such as those that are zero or close to zero Albericio et al. (2016a). Minerva is a DNN hardware generator which also takes advantage of zero activations and that targets high-energy efficiency Reagen et al. (2016). Layer fusion can further reduce off-chip communication and create additional parallelism Alwani et al. (2016). As multiple layers are processed concurrently, a straightforward combination with *TRT* would use the maximum of the precisions when layers are fused.

Google's Tensor Processing Unit uses quantization to represent values using 8 bits Jouppi (2016) to support TensorFlow Abadi et al. (2015). As Table 1 shows, some layers can use lower than 8 bits of precision which suggests that even with quantization it may be possible to use fewer levels and to potentially benefit from an engine such as *TRT*.

**Limitations:** As in *DaDN* this work assumed that each layer fits on-chip. However, as networks evolve it is likely that they will increase in size thus requiring multiple *TRT* nodes as was suggested in *DaDN*. However, some newer networks tend to use more but smaller layers. Regardless, it would be desirable to reduce the area cost of *TRT* most of which is due to the eDRAM buffers. We have not explored this possibility in this work. Proteus Judd et al. (2016b) is directly compatible with *TRT* and can reduce memory footprint by about 60% for both convolutional and fully-connected layers. Ideally, compression, quantization and pruning similar in spirit to EIE Han et al. (2016) would be used to reduce computation, communication and footprint. General memory compresion Mittal & Vetter (2016) techniques offer additional opportunities for reducing footprint and communication.

We evaluated *TRT* only on CNNs for image classification. Other network architectures are important and the layer configurations and their relative importance varies. *TRT* enables performance

improvements for two of the most dominant layer types. We have also provided some preliminary evidence that *TRT* works well for NeuralTalk LSTM Karpathy & Li (2014). Moreover, by enabling output activation computation slicing it can accommodate relatively small layers as well.

Applying some of the concepts that underlie the *TRT* design to other more general purpose accelerators such as Cambricon Liu et al. (2016) or graphics processors would certainly be more preferable than a dedicated accelerator in most application scenarios. However, these techniques are best first investigated into specific designs and then can be generalized appropriately.

We have evaluated *TRT* only for inference only. Using an engine whose performance scales with precision would provide another degree of freedom for network training as well. However, *TRT* needs to be modified accordingly to support all the operations necessary during training and the training algorithms need to be modified to take advantage of precision adjustments.

This section commented only on related work on digital hardware accelerators for DNNs. Advances at the algorithmic level would impact *TRT* as well or may even render it obsolete. For example, work on using binary weights Courbariaux et al. (2015) would obviate the need for an accelerator whose performance scales with weight precision. Investigating *TRT*'s interaction with other network types and architectures and other machine learning algorithms is left for future work.

## 6 CONCLUSION

This work presented *Tartan* an accelerator for inference with Deep Learning Networks whose performance scales inversely linearly with the number of bits used to represent values in fully-connected and convolutional layers. *TRT* also enables on-the-fly accuracy vs. performance and energy efficiency trade offs and its benefits were demonstrated over a set of popular image classification networks. The new key ideas in *TRT* are: 1) Supporting both the bit-parallel and the bit-serial loading of weights into processing units to facilitate the processing of either convolutional or fully-connected layers, and 2) cascading the adder trees of various subunits (SIPs) to enable slicing the output computation thus reducing or eliminating cross-lane imbalance for relatively small layers.

*TRT* opens up a new direction for research in inference and training by enabling precision adjustments to translate into performance and energy savings. These precisions adjustments can be done statically prior to execution or dynamically during execution. While we demonstrated *TRT* for inference only, we believe that *TRT*, especially if combined with Pragmatic, opens up a new direction for research in training as well. For systems level research and development, *TRT* with its ability to trade off accuracy for performance and energy efficiency enables a new degree of adaptivity for operating systems and applications.

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
