# Peer review of "Tartan: Accelerating Fully-Connected and Convolutional Layers in Deep Learning Networks by Exploiting Numerical Precision Variability"

_ICLR 2017 — rejected_

[Official Review · AnonReviewer2 · rating 4 · confidence 1 · 18 Dec 2016]
**My thoughts too**

I do not feel very qualified to review this paper. I studied digital logic back in university, that was it. I think the work deserves a reviewer with far more sophisticated background in this area. It certainly seems useful. My advice is also to submit it another venue.

[Official Review · AnonReviewer3 · rating 4 · confidence 3 · 18 Dec 2016]
**consider a better venue for submission**

This paper proposed a hardware accelerator for DNN. It utilized the fact that DNN are very tolerant to low precision inference and outperforms a state-of-the-art bit-parallel accelerator by 1.90x without any loss in accuracy while it is 1.17x more energy efficient. TRT requires no network retraining. It achieved super linear scales of performance with area.

The first concern is that this paper doesn't seem very well-suited to ICLR. The circuit diagrams makes it more interesting for the hardware or circuit design community. 

The second concern is the "take-away for machine learning community", seeing from the response, the take-away is using low-precision to make inference cheaper. This is not novel enough. In last year's ICLR, there were at least 4 papers discussing using low precision to make DNN more efficient. These ideas have also been explored in the authors' previous papers.

[Public Comment · ICLR 2017 conference · 19 Dec 2016]
**hardware as a topic for ICLR**

Hardware is listed on the call-for-papers as relevant topic for ICLR 2017, and so the paper is on-topic.

We worked hard to improve on the initial reviewer assignment for this paper in ensure that we would get hardware-knowledgeable reviewers on board for this paper, although we only partly succeeded (due to conflicts, tight review deadlines, and more).

We are also still missing questions and comments from one reviewer.

ICLR papers should succeed in communicating with the ICLR audience, i.e., introducing concepts for this audience, and using a language that is accessible for this community.  But it need not connect with all of the ICLR community, which is dominated by algorithmic concerns; many may not have much background or interest in hardware and that is fine.  This situation is not unique, e.g., hardware-related papers in computer graphics and computer vision conferences are in the same situation.  It is also true, I think, that the audience and reviewers of hardware-related conferences may not be well-placed to fully understand and comment on all relevant algorithmic considerations that a hardware implementation targets. 

There remains disagreement as to the utility of achieving improvements related to fully connected layers, which would be good to resolve. And there also remains disagreement on the improvements (conceptual and performance-related) with respect to the state of the art, as recently rebutted by the authors.

[Official Review · AnonReviewer5 · rating 5 · confidence 5 · 21 Dec 2016]
**Stripes and Tartan are interesting architectures but the contribution over the three previous publications on this idea is extremely small**

Summary:

The paper describes how the DaDianNao (DaDN) DNN accelerator can be improved by employing bit serial arithmetic.  They replace the bit-parallel multipliers in DaDN with multipliers that accept the weights in parallel but the activations serially (serial x parallel multipliers).  They increase the number of units keeping the total number of adders constant.  This enables them to tailor the time and energy consumed to the number of bits used to represent activations.  They show how their configuration can be used to process both fully-connected and convolutional layers of DNNs.

Strengths:

Using variable precision for each layer of the network is useful - but was previously reported in Judd (2015)

Good evaluation including synthesis - but not place and route - of the units.  Also this evaluation is identical to that in Judd (2016b)

Weaknesses:

The idea of combining bit-serial arithmetic with the DaDN architecture is a small one.

The authors have already published almost everything that is in this paper at Micro 2016 in Judd (2016b).  The increment here is the analysis of the architecture on fully-connected layers.  Everything else is in the previous publication.

The energy gains are small - because the additional flip-flop energy of shifting the activations in almost offsets the energy saved on reducing the precision of the arithmetic.

The authors don’t compare to more conventional approaches to variable precision - using bit-parallel arithmetic units but data gating the LSBs so that only the relevant portion of the arithmetic units toggle.  This would not provide any speedup, but would likely provide better energy gains than the bit-serial x bit-parallel approach.

Overall:

The Tartan and Stripes architectures are interesting but the incremental contribution of this paper (adding support for fully-connected layers) over the three previous publications on this topic, and in particular Judd (2016b) is very small.  This idea is worth one good paper, not four.

[Official Review · AnonReviewer1 · rating 6 · confidence 2 · 21 Dec 2016]
**Improving inference speed and energy-efficiency in (simulated) hardware implementations by exploiting per-layer differences in numerical precision requirements.**

This seems like a reasonable study, though it's not my area of expertise. I found no fault with the work or presentation, but did not follow the details or know the comparable literature. 

There seem to be real gains to be had through this technique, though they are only in terms of efficiency in hardware, not changing accuracy on a task. The tasks chosen (Alexnet / VGG) seem reasonable. The results are in simulation rather than in actual hardware.

The topic seems a little specialized for ICLR, since it does not describe any new advances in learning or representations, albeit that the CFP includes "hardware".  I think the appeal among attendees will be rather limited. 

Please learn to use parenthetical references correctly. As is your references make reading harder.

[Official Review · AnonReviewer4 · rating 5 · confidence 5 · 29 Dec 2016]
**Incremental, perhaps better suited for an architecture conference (ISCA/ ASPLOS)**

The authors present TARTAN, a derivative of the previously published DNN accelerator architecture: “DaDianNao”. The key difference is that TARTAN’s compute units are bit-serial and unroll MAC operation over several cycles. This enables the units to better exploit any reduction in precision of the input activations for improvement in performance and energy efficiency.

Comments:

1. I second the earlier review requesting the authors to be present more details on the methodology used for estimating energy numbers for TARTAN. It is claimed that TARTAN gives only a 17% improvement in energy efficiency. However, I suspect that this small improvement is clearly within the margin of error ij energy estimation.  

2. TARTAN is a derivative of DaDianNao, and it heavily relies the overall architecture of DaDianNao. The only novel aspect of this contribution is the introduction of the bit-serial compute unit, which (unfortunately) turns out to incur a severe area overhead (of nearly 3x over DaDianNao's compute units).

3. Nonetheless, the idea of bit-serial computation is certainly quite interesting. I am of the opinion that it would be better appreciated (and perhaps be even more relevant) in a circuit design / architecture focused venue.

[Author Response · Andreas Moshovos · 20 Jan 2017]
**Reviewer Rating**

We feel uncomfortable completing the reviewer ratings as provided except for one case where a review misrepresents the facts and which we have addressed. This is because the reviews primarily state that the paper does not fit into ICLR. This is a question for the organizers and an interpretation of the CFP which clearly states "hardware".

In summary, the take-away for the ML community is this:

1. Adjusting the precision used per layer or even at a finer granularity of groups of 256 or activations and weights can lead to faster processing and higher energy efficiency. This adjustment can be done at runtime and at a single-bit granularity. Performance can be had without sacrificing accuracy, but if one is willing to sacrifice accuracy more performance and more energy efficiency can be had. We envision follow up work on runtime adjustment of precisions (e.g,, incremental adjustment) to achieve better response times or higher throughput.

In more detail:

2. There is an energy efficient high-performance hardware design that can offer performance inversely proportional to the precision being used per layer or even at a finer granularity (we do not present any results on this but the design obviously support it as-is). 

2. This design does not hardwire the precisions at manufacturing time but instead allows programmatic control at runtime.

3. This capability opens up an additional design know that network designers can use to tradeoff execution time and accuracy.

4. Earlier we had proposed a method to choose per layer precisions  for convolutional layers here we extend this method to fully-connected layers. This method was published on arxiv and has never been accepted to any peer reviewed publication. The only related publication to the work here is STRIPES (MICRO) (12 pages) and a pre-print at iEEE Computer Architecture Letters (4 pages) that explained the basic idea behind Stripes. This work extends STRIPES for Fully-connected layers --- Stripes did not improve performance for FC layers and its energy efficiency was worse than DaDianNao for those layers.

Concern summary:

1. Some concerns were raised about the energy and area measurements. We have posted an update and we will deliver the final results post-layout in a day or so. We will also deliver results on an optimized configuration that drastically reduces area overhead and improves energy efficiency as well. The delay in response was due to having received the reviews during the Xmas break when the author that can perform these measurements was unreachable due to travel. 

2. Incremental over DaDianNao: Tartan is a general concept which can be integrated to many different architectures. We chose DaDN as the baseline architecture as it widely known and often compared against and offers the additional challenge of being a very wide vector-like architecture (doing bit-serial computation for a single lane -- product -- independently is easy but doing it for 4K terms in parallel without extremely wide memories is hard). So, we disagree that this is an incremental improvement of DaDianNao. To draw an analogy, from hardware the seminal work on pattern based branch prediction was not an incremental improvement over out-of-order execution even though previous techniques included branch predictors. Not that we feel that TARTAN is at the same level as pattern based branch prediction but we use this example to illustrate what such an argument can lead to. Also, STRIPES is receiving a honorable mention in the upcoming IEEE MICRO Topic PIcs in Computer Architecture researcher, which is akin to a best paper award in the field of computer architecture (each accepted paper in the last year in a top-tier conference receives 10 more peer reviews that state whether they feel the paper has the potential for high impact). So, at least some people that are credible enough to be invited to that panel in the comp arch community think it's not incremental.

3. FC layers are not important: this is not true. They are still in use and more so in different applications. Moreover, last years best paper award in ICLR rightfully went to an excellent work that addressed both pruning the model and proposing an optimized hardware architecture solely for FC layers.


4. TARTAN takes too much area: The units are larger but this is not where most of the area cost is. The area cost is in the surrounding memory so in the big picture the overall area cost is much lower. also, in modern technology area is not the concern.

5. Energy efficiency is not that much better than DaDN and within the error margin of the tools. We used industry standard tools and the results are positive. moreover, TARTAN is faster and has we can use frequency and voltage scaling to improve energy efficiency (which depends on voltage square) while still performing better. We show results assuming same voltage and frequency.

6. You did not use power gating for DaDN. Did we not use it for TARTAN either. It is not straightforward to do for DaDN as it is a bit parlallel engine and there are not that many zero values to necessarily justify the logic needed to enable power gating. Which is to say that this ia a non-trivial task. Moreover, the same technique can be applied to TATRAN. We have results reported in another submission that show a heavy bias toward the zero bit value which suggestes that power gating will most likely be a lot more effective for TARTAN than DaDN.

In summary, combined this work, with STRIPES and the earlier arxiv report present a comprehensive hardware/software approach to exploiting precision to improve performance and energy efficiency for CNNs. Stripes performed very well for convolutional layers but did poorly for fully-connected layers. TARTAN fixes this.

[Author Response · Andreas Moshovos · 20 Jan 2017]
**Updated latency and energy efficiency measurements**

Our apologies for the long delay. Our co-author that has the expertise to do this work was overseas for the Christmas break and she returned this Tuesday.

She has synthesized the designs for three cases: bc, tc, and wc for best, typical and worst case respectively. The previous results were for the bc. The detailed results are below. In summary, efficiency improves for tc and wc. We have layout being synthesized and we expect to have the results in a day or so. The results below are pre-layout and use 50% activity factors. The layout results that we will post as soon as possible will be a testbench for a typical layer. As with STRIPES we expect that energy efficiency will be better with real inputs as the inputs exhibit many more zero bits. Please keep in mind that the results in the STRIPES publication in MICRO are post-layout.

More importantly, we are also synthesizing a different configuration cuts down area costs and improves energy efficiency even further. We will post the post-layout results shortly.

We hope that we will be given the opportunity to post the updated results before a decision is made.

Thank you.

Here are the detailed pre-layout results for all three cases and for fully-connected layers only. Best case is the configuration used in the original submission. We will update the writing with post-layout and actual activity-based results shortly. Effieciency numbers above 1.0 mean that TRT is better than DaDN.

Area & Efficiency, Fully-connected layers, whole chip TRT vs DaDN:

Best case

Baseline area - 77.91 mm2
TRT area - 108.61 mm2 (+39.4%)
Efficiency:
AlexNet  0.935
VGG_S    0.932
VGG_M    0.935
VGG_19   0.932
geomean  0.933

Typical case

Baseline area - 79.28 mm2
TRT area - 111.29 mm2 (+40.4%)
Efficiency:
AlexNet  1.014
VGG_S    1.011
VGG_M    1.014
VGG_19   1.010
geomean  1.012

Worst case

Baseline area - 83.5 mm2
TRT area - 121.39 mm2 (+45.3%)
Efficiency:
AlexNet  1.048
VGG_S    1.046
VGG_M    1.049
VGG_19   1.045
geomean  1.047

[Author Response · Andreas Moshovos · 21 Jan 2017]
**Post-Layout Results and a better configuration**

Here are our post-layout results with actual data-driven activity factors. The 2-bit configuration, detailed in the updated paper, is a design that processes two bits at once using half as many SIPs, increasing efficiency and reducing area overhead. It requires that precision is an even number.

We report results for the typical design case. Pre-layout results showed that the worst case design corner results in a larger advantage for TRT. The results show increased energy efficiency compared to the pre-layout results. This is expected as the pre-layout results used 50% activity factors whereas here we use actual activity factors measured on a typical layer. This is consistent with the behavior observed during the STRIPES work.

Fully-connected layers, whole chip (TSMC 65nm typical case):

Baseline (DaDN) area - 80.41 mm2

TRT area - 120.04 mm2
Efficiency (TRT vs baseline):
AlexNet  1.062
VGG_S    1.059
VGG_M    1.063
VGG_19   1.059
geomean  1.061

TRT 2-bit area - 100.43 mm2
Efficiency (TRT 2-bit vs baseline):
AlexNet  1.228
VGG_S    1.235
VGG_M    1.268
VGG_19   1.237
geomean  1.242

Numbers greater than 1 mean less energy used overall by TRT.

[Final Decision · Program Chairs · 06 Feb 2017]
**ICLR committee final decision**

This metareview is written based on the reviews of the two expert reviewers (scores of 5(5), 5(5)),
 given the wide disparity of reviewer expertise, and the relevance of hardware expertise to this particular paper.
 
 Quality, Clarity: The paper is clearly written. It does require hardware expertise to read and appreciate the ideas, methodology, and the results. There was a request to post a clear "take home" message for ML research, and the authors did post a clear statement in this regard.
 
 Originality: The size of the contribution is the major point of contention for this paper. The expert reviewers believe it to be an incremental/small idea, particularly in relation to the prior work of the authors. The authors provide rebut that while the implementation builds on their previous work, it can be seen as a general concept that could be applied to other architectures, and that this work targets the important case of fully-connected architectures, vs the case of convolutional architectures that were the case where the prior work did well.
 
 Overall: I have asked the expert reviewers for further input. In the absence of further information, updated scores, or some reasonably strong indication of excitement about the ideas from the expert reviewers, there is little choice but to currently lean towards "reject" in terms of this being an inspirational paper for ICLR.